# Minimizing FLOPs to Learn Efficient Sparse Representations

**Biswajit Paria**[†*], **Chih-Kuan Yeh**[†], **Ian E.H. Yen**[‡], **Ning Xu**[§], **Pradeep Ravikumar**[†], **Barnabás Póczos**[†]

[†]Carnegie Mellon University, [‡]Moffett AI, [§]Amazon

{bparia,cjyeh,pradeepr,bapoczos}@cs.cmu.edu, ian.yan@moffett.ai, ningxu01@gmail.com

## Abstract

Deep representation learning has become one of the most widely adopted approaches for visual search, recommendation, and identification. Retrieval of such representations from a large database is however computationally challenging. Approximate methods based on learning compact representations, have been widely explored for this problem, such as *locality sensitive hashing*, *product quantization*, and *PCA*. In this work, in contrast to learning compact representations, we propose to learn high dimensional and sparse representations that have similar representational capacity as dense embeddings while being more efficient due to sparse matrix multiplication operations which can be much faster than dense multiplication. Following the key insight that the number of operations decreases quadratically with the sparsity of embeddings provided the non-zero entries are distributed uniformly across dimensions, we propose a novel approach to learn such distributed sparse embeddings via the use of a carefully constructed regularization function that directly minimizes a continuous relaxation of the number of floating-point operations (FLOPs) incurred during retrieval. Our experiments show that our approach is competitive to the other baselines and yields a similar or better speed-vs-accuracy tradeoff on practical datasets[1].

## 1 Introduction

Learning semantic representations using deep neural networks (DNN) is now a fundamental facet of applications ranging from visual search (Jing et al., 2015; Hadi Kiapour et al., 2015), semantic text matching (Neculoiu et al., 2016), oneshot classification (Koch et al., 2015), clustering (Oh Song et al., 2017), and recommendation (Shankar et al., 2017). The high-dimensional dense embeddings generated from DNNs however pose a computational challenge for performing nearest neighbor search in large-scale problems with millions of instances. In particular, when the embedding dimension is high, evaluating the distance of any query to all the instances in a large database is expensive, so that efficient search without sacrificing accuracy is difficult. Representations generated using DNNs typically have a higher dimension compared to hand-crafted features such as SIFT (Lowe, 2004), and moreover are dense. The key caveat with dense features is that unlike bag-of-words features they cannot be efficiently searched through an inverted index, without approximations.

Since accurate search in high dimensions is prohibitively expensive in practice (Wang, 2011), one has to typically sacrifice accuracy for efficiency by resorting to approximate methods. Addressing the problem of efficient approximate *Nearest-Neighbor Search (NNS)* (Jegou et al., 2011) or *Maximum Inner-Product Search (MIPS)* (Shrivastava and Li, 2014) is thus an active area of research, which we review in brief in the related work section. Most approaches (Charikar, 2002; Jegou et al., 2011) aim to learn compact lower-dimensional representations that preserve distance information.

While there has been ample work on learning compact representations, learning sparse higher dimensional representations have been addressed only recently (Jeong and Song, 2018; Cao et al., 2018). As a seminal instance, Jeong and Song (2018) propose an end-to-end approach to learn

---

[*]Part of the work was done when BP was a research intern at Snap Inc.
[1]The implementation is available at https://github.com/biswajitsc/sparse-embed

sparse and high-dimensional hashes, showing significant speed-up in retrieval time on benchmark datasets compared to dense embeddings. This approach has also been motivated from a biological viewpoint (Li et al., 2018) by relating to a fruit fly's olfactory circuit, thus suggesting the possibility of hashing using higher dimensions instead of reducing the dimensionality. Furthermore, as suggested by Glorot et al. (2011), sparsity can have additional advantages of linear separability and information disentanglement.

In a similar vein, in this work, we propose to learn high dimensional embeddings that are sparse and hence efficient to retrieve using sparse matrix multiplication operations. In contrast to compact lower-dimensional ANN-esque representations that typically lead to decreased representational power, a key facet of our higher dimensional sparse embeddings is that they can have the same representational capacity as the initial dense embeddings. The core idea behind our approach is inspired by two key observations: (i) retrieval of $d$ (high) dimensional sparse embeddings with fraction $p$ of non-zero values on an average, can be sped up by a factor of $1/p$. (ii) The speed up can be further improved to a factor of $1/p^2$ by ensuring that the non-zero values are evenly distributed across all the dimensions. This indicates that sparsity alone is not sufficient to ensure maximal speedup; the distribution of the non-zero values plays a significant role as well. This motivates us to consider the effect of sparsity on the number of *floating point operations (FLOPs)* required for retrieval with an inverted index. We propose a penalty function on the embedding vectors that is a continuous relaxation of the exact number of FLOPs, and encourages an even distribution of the non-zeros across the dimensions.

We apply our approach to the large scale metric learning problem of learning embeddings for facial images. Our training loss consists of a *metric learning* (Weinberger and Saul, 2009) loss aimed at learning embeddings that mimic a desired metric, and a *FLOPs loss* to minimize the number of operations. We perform an empirical evaluation of our approach on the Megaface dataset (Kemelmacher-Shlizerman et al., 2016), and show that our proposed method successfully learns high-dimensional sparse embeddings that are orders-of-magnitude faster. We compare our approach to multiple baselines demonstrating an improved or similar speed-vs-accuracy trade-off.

The rest of the paper is organized as follows. In Section 3 we analyze the expected number of FLOPs, for which we derive an exact expression. In Section 4 we derive a continuous relaxation that can be used as a regularizer, and optimized using gradient descent. We also provide some analytical justifications for our relaxation. In Section 5 we then compare our method on a large metric learning task showing an improved speed-accuracy trade-off compared to the baselines.

## 2 Related Work

**Learning compact representations, ANN.** Exact retrieval of the top-k nearest neighbours is expensive in practice for high-dimensional dense embeddings learned from deep neural networks, with practitioners often resorting to *approximate nearest neighbours* (ANN) for efficient retrieval. Popular approaches for ANN include *Locality sensitive hashing* (LSH) (Gionis et al., 1999; Andoni et al., 2015; Raginsky and Lazebnik, 2009) relying on random projections, *Navigable small world graphs* (NSW) (Malkov et al., 2014) and hierarchical NSW (HNSW) (Malkov and Yashunin, 2018) based on constructing efficient search graphs by finding clusters in the data, *Product Quantization* (PQ) (Ge et al., 2013; Jegou et al., 2011) approaches which decompose the original space into a cartesian product of low-dimensional subspaces and quantize each of them separately, and *Spectral hashing* (Weiss et al., 2009) which involves an NP hard problem of computing an optimal binary hash, which is relaxed to continuous valued hashes, admitting a simple solution in terms of the spectrum of the similarity matrix. Overall, for compact representations and to speed up query times, most of these approaches use a variety of carefully chosen data structures, such as hashes (Neyshabur and Srebro, 2015; Wang et al., 2018), locality sensitive hashes (Andoni et al., 2015), inverted file structure (Jegou et al., 2011; Baranchuk et al., 2018), trees (Ram and Gray, 2012), clustering (Auvolat et al., 2015), quantization sketches (Jegou et al., 2011; Ning et al., 2016), as well as dimensionality reductions based on principal component analysis and t-SNE (Maaten and Hinton, 2008).

**End to end ANN.** Learning the ANN structure end-to-end is another thread of work that has gained popularity recently. Norouzi et al. (2012) propose to learn binary representations for the Hamming metric by minimizing a margin based triplet loss. Erin Liong et al. (2015) use the signed output of a deep neural network as hashes, while imposing independence and orthogonality conditions on the

hash bits. Other end-to-end learning approaches for learning hashes include (Cao et al., 2016; Li et al., 2017). An advantage of end-to-end methods is that they learn hash codes that are optimally compatible to the feature representations.

**Sparse representations.**   Sparse representations have been previously studied from various viewpoints. Glorot et al. (2011) explore sparse neural networks in modeling biological neural networks and show improved performance, along with additional advantages such as better linear separability and information disentangling. Ranzato et al. (2008; 2007); Lee et al. (2008) propose learning sparse features using deep belief networks. Olshausen and Field (1997) explore sparse coding with an overcomplete basis, from a neurobiological viewpoint. Sparsity in auto-encoders have been explored by Ng et al. (2011); Kavukcuoglu et al. (2010). Arpit et al. (2015) provide sufficient conditions to learn sparse representations, and also further provide an excellent review of sparse autoencoders. Dropout (Srivastava et al., 2014) and a number of its variants (Molchanov et al., 2017; Park et al., 2018; Ba and Frey, 2013) have also been shown to impose sparsity in neural networks.

**High-dimensional sparse representations.**   *Sparse deep hashing (SDH)* (Jeong and Song, 2018) is an end-to-end approach that involves starting with a pre-trained network and then performing alternate minimization consisting of two minimization steps, one for training the binary hashes and the other for training the continuous dense embeddings. The first involves computing an optimal hash best compatible with the dense embedding using a *min-cost-max-flow* approach. The second step is a gradient descent step to learn a dense embedding by minimizing a metric learning loss. A related approach, $k$-sparse autoencoders (Makhzani and Frey, 2013) learn representations in an unsupervised manner with at most $k$ non-zero activations. The idea of high dimensional sparse embeddings is also reinforced by the *sparse-lifting* approach (Li et al., 2018) where sparse high dimensional embeddings are learned from dense features. The idea is motivated by the biologically inspired *fly* algorithm (Dasgupta et al., 2017). Experimental results indicated that *sparse-lifting* is an improvement both in terms of precision and speed, when compared to traditional techniques like LSH that rely on dimensionality reduction.

$\ell_1$ **regularization, lasso.**   The *Lasso* (Tibshirani, 1996) is the most popular approach to impose sparsity and has been used in a variety of applications including sparsifying and compressing neural networks (Liu et al., 2015; Wen et al., 2016). The *group lasso* (Meier et al., 2008) is an extension of lasso that encourages all features in a specified group to be selected together. Another extension, the *exclusive lasso* (Kong et al., 2014; Zhou et al., 2010), on the other hand, is designed to select a single feature in a group. Our proposed regularizer, originally motivated by idea of minimizing FLOPs closely resembles exclusive lasso. Our focus however is on sparsifying the produced embeddings rather than sparsifying the parameters.

**Sparse matrix vector product (SpMV).**   Existing work for SpMV computations include (Haffner, 2006; Kudo and Matsumoto, 2003), proposing algorithms based on inverted indices. Inverted indices are however known to suffer from severe cache misses. Linear algebra back-ends such as BLAS (Blackford et al., 2002) rely on efficient cache accesses to achieve speedup. Haffner (2006); Mellor-Crummey and Garvin (2004); Krotkiewski and Dabrowski (2010) propose cache efficient algorithms for sparse matrix vector products. There has also been substantial interest in speeding up SpMV computations using specialized hardware such as GPUs (Vazquez et al., 2010; Vázquez et al., 2011), FPGAs (Zhuo and Prasanna, 2005; Zhang et al., 2009), and custom hardware (Prasanna and Morris, 2007).

**Metric learning.**   While there exist many settings for learning embeddings (Hinton and Salakhutdinov, 2006; Kingma and Welling, 2013; Kiela and Bottou, 2014) in this paper we restrict our attention to the context of metric learning (Weinberger and Saul, 2009). Some examples of metric learning losses include large margin softmax loss for CNNs (Liu et al., 2016), triplet loss (Schroff et al., 2015), and proxy based metric loss (Movshovitz-Attias et al., 2017).

## 3   EXPECTED NUMBER OF FLOPS

In this section we study the effect of sparsity on the expected number of FLOPs required for retrieval and derive an exact expression for the expected number of FLOPs. The main idea in this paper

is based on the key insight that if each of the dimensions of the embedding are non-zero with a probability $p$ (not necessarily independently), then it is possible to achieve a speedup up to an order of $1/p^2$ using an inverted index on the set of embeddings. Consider two embedding vectors $\boldsymbol{u}, \boldsymbol{v}$. Computing $\boldsymbol{u}^T \boldsymbol{v}$ requires computing only the pointwise product at the indices $k$ where both $\boldsymbol{u}_k$ and $\boldsymbol{v}_k$ are non-zero. This is the main motivation behind using inverted indices and leads to the aforementioned speedup. Before we analyze it more formally, we introduce some notation.

Let $\mathcal{D} = \{(\boldsymbol{x}_i, y_i)\}_{i=1}^n$ be a set of $n$ independent training samples drawn from $\mathcal{Z} = \mathcal{X} \times \mathcal{Y}$ according to a distribution $\mathcal{P}$, where $\mathcal{X}, \mathcal{Y}$ denote the input and label spaces respectively. Let $\mathscr{F} = \{f_\theta : \mathcal{X} \to \mathbb{R}^d \mid \theta \in \Theta\}$ be a class of functions parameterized by $\theta \in \Theta$, mapping input instances to $d$-dimensional embeddings. Typically, for image tasks, the function is chosen to be a suitable CNN (Krizhevsky et al., 2012). Suppose $X, Y \sim \mathcal{P}$, then define the activation probability $p_j = \mathbb{P}(f_\theta(X)_j \neq 0)$, and its empirical version $\bar{p}_j = \frac{1}{n} \sum_{i=1}^n \mathbb{I}[f_\theta(\boldsymbol{x}_i)_j \neq 0]$.

We now show that sparse embeddings can lead to a quadratic speedup. Consider a $d$-dimensional sparse query vector $\boldsymbol{u}_q = f_\theta(\boldsymbol{x}_q) \in \mathbb{R}^d$ and a database of $n$ sparse vectors $\{\boldsymbol{v}_i = f_\theta(\boldsymbol{x}^{(i)})\}_{i=1}^n \subset \mathbb{R}^d$ forming a matrix $\boldsymbol{D} \in \mathbb{R}^{n \times d}$. We assume that $\boldsymbol{x}_q, \boldsymbol{x}^{(i)}$ $(i = 1, \ldots, n)$ are sampled independently from $\mathcal{P}$. Computing the vector matrix product $\boldsymbol{D}\boldsymbol{u}_q$ requires looking at only the columns of $\boldsymbol{D}$ corresponding to the non-zero entries of $\boldsymbol{u}_q$ given by $N_q = \{j \mid 1 \leq j \leq d, (\boldsymbol{u}_q)_j \neq 0\}$. Furthermore, in each of those columns we only need to look at the non-zero entries. This can be implemented efficiently in practice by storing the non-zero indices for each column in independent lists, as depicted in Figure 2.

The number of FLOPs incurred is given by,

$$F(\boldsymbol{D}, \boldsymbol{u}_q) = \sum_{j \in N_q} \sum_{i: \boldsymbol{v}_{ij} \neq 0} 1 = \sum_{i=1}^n \sum_{j=1}^d \mathbb{I}[(\boldsymbol{u}_q)_j \neq 0 \wedge \boldsymbol{v}_{ij} \neq 0]$$

Taking the expectation on both sides w.r.t. $\boldsymbol{x}_q, \boldsymbol{x}^{(i)}$ and using the independence of the data, we get

$$\mathbb{E}[F(\boldsymbol{D}, \boldsymbol{u}_q)] = \sum_{i=1}^n \sum_{j=1}^d \mathbb{P}\big((\boldsymbol{u}_q)_j \neq 0\big) \mathbb{P}\big(\boldsymbol{v}_{ij} \neq 0\big) = n \sum_{j=1}^d \mathbb{P}(f_\theta(X)_j \neq 0)^2 \qquad (1)$$

where $X \sim \mathcal{P}$. Since the expected number of FLOPs scales linearly with the number of vectors in the database, a more suitable quantity is the *mean-FLOPs-per-row* defined as

$$\mathcal{F}(f_\theta, \mathcal{P}) = \mathbb{E}[F(\boldsymbol{D}, \boldsymbol{u}_q)]/n = \sum_{j=1}^d \mathbb{P}(f_\theta(X)_j \neq 0)^2 = \sum_{j=1}^d p_j^2. \qquad (2)$$

Note that for a fixed amount of sparsity $\sum_{j=1}^d p_j = d\, p$, this is minimized when each of the dimensions are non-zero with equal probability $p_j = p, \forall 1 \leq j \leq d$, upon which $\mathcal{F}(f_\theta, \mathcal{P}) = d\, p^2$ (so that as a regularizer, $\mathcal{F}(f_\theta, \mathcal{P})$ will in turn encourage such a uniform distribution across dimensions). Given such a uniform distribution, compared to dense multiplication which has a complexity of $O(d)$ per row, we thus get an improvement by a factor of $1/p^2$ $(p < 1)$. Thus when only $p$ fraction of all the entries is non-zero, and evenly distributed across all the columns, we achieve a speedup of $1/p^2$. Note that independence of the non-zero indices is not necessary due to the linearity of expectation – in fact, features from a neural network are rarely uncorrelated in practice.

**FLOPs versus speedup.** While FLOPs reduction is a reasonable measure of speedup on primitive processors of limited parallelization and cache memory. FLOPs is not an accurate measure of actual speedup when it comes to mainstream commercial processors such as Intel's CPUs and Nvidia's GPUs, as the latter have cache and SIMD (Single-Instruction Multiple Data) mechanism highly optimized for dense matrix multiplication, while sparse matrix multiplication are inherently less tailored to their cache and SIMD design (Sohoni et al., 2019). On the other hand, there have been threads of research on hardwares with cache and parallelization tailored to sparse operations that show speedup proportional to the FLOPs reduction (Han et al., 2016; Parashar et al., 2017). Modeling the cache and other hardware aspects can potentially lead to better performance but less generality and is left to our future works.

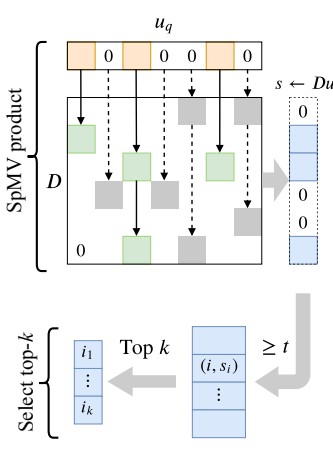

**Algorithm 1** Sparse Nearest Neighbour

1: *(Build Index)*
2: **Input:** Sparse matrix $D$
3:   **for** $j = 1 \cdots d$ **do**
4:     Init $C[j] \leftarrow \{(i, D_{ij}) \mid D_{ij} \neq 0 \ \wedge \ 1 \leq i \leq n\}$
5:       $\triangleright$ Stores the non-zero values and their indices
6:   **end for**
7:
8: *(Query)*
9: **Input:** Sparse query $u_q$, threshold $t$, number of NNs $k$
10:   Init score vector $s[i] = 0, \ 1 \leq i \leq n$.
11:   **for** $j = 1 \cdots d$ s.t. $u_q[j] \neq 0$ **do**      $\triangleright$ SpMV product
12:     **for** $(i, v) \in C[j]$ **do**
13:       $s[i] \mathrel{+}= v u_q[j]$
14:     **end for**
15:   **end for**
16:   $S \leftarrow \{(i, s[i]) \mid 1 \leq i \leq n, \ s[i] \geq t\}$      $\triangleright$ Thresholding
17:   $S_k \leftarrow \{i_1, \ldots, i_k\}$ top-$k$ indices $i$ of $S$ based on $s[i]$
18:         $\triangleright$ Using **nth_select** from C++ STL
19:   **return** $S_k$

Figure 2: **SpMV product:** The colored cells denote non-zero entries, and the arrows indicate the list structure for each of the columns, with solid arrows denoting links that were traversed for the given query. The green and grey cells denote the non-zero entries that were accessed and not accessed, respectively. The non-zero values in $Du_q$ (blue) can be computed using only the common non-zero values (green). **Selecting top-$k$:** The sparse product vector is then filtered using a threshold $t$, after which the top-$k$ indices are returned.

## 4 Our Approach

The $\ell_1$ regularization is the most common approach to induce sparsity. However, as we will also verify experimentally, it does not ensure an uniform distribution of the non-zeros in all the dimensions that is required for the optimal speed-up. Therefore, we resort to incorporating the actual FLOPs incurred, directly into the loss function which will lead to an optimal trade-off between the search time and accuracy. The FLOPs $\mathcal{F}(f_\theta, \mathcal{P})$ being a discontinuous function of model parameters, is hard to optimize, and hence we will instead optimize using a continuous relaxation of it.

Denote by $\ell(f_\theta, \mathcal{D})$, any metric loss on $\mathcal{D}$ for the embedding function $f_\theta$. The goal in this paper is to minimize the loss while controlling the expected FLOPs $\mathcal{F}(f_\theta, \mathcal{P})$ defined in Eqn. 2. Since the distribution $\mathcal{P}$ is unknown, we use the samples to get an estimate of $\mathcal{F}(f_\theta, \mathcal{P})$. Recall the empirical fraction of non-zero activations $\bar{p}_j = \frac{1}{n} \sum_{i=1}^{n} \mathbb{I}[f_\theta(x_i)_j \neq 0]$, which converges in probability to $p_j$. Therefore, with a slight abuse of notation define $\mathcal{F}(f_\theta, \mathcal{D}) = \sum_{j=1}^{d} \bar{p}_j^2$, which is a consistent estimator for $\mathcal{F}(f_\theta, \mathcal{P})$ based on the samples $\mathcal{D}$. Note that $\mathcal{F}$ denotes either the population or empirical quantities depending on whether the functional argument is $\mathcal{P}$ or $\mathcal{D}$. We now consider the following regularized loss.

$$\min_{\theta \in \Theta} \ \underbrace{\ell(f_\theta, \mathcal{D}) + \lambda \mathcal{F}(f_\theta, \mathcal{D})}_{\mathcal{L}(\theta)} \tag{3}$$

for some parameter $\lambda$ that controls the FLOPs-accuracy tradeoff. The regularized loss poses a further hurdle, as $\bar{p}_j$ and consequently $\mathcal{F}(f_\theta, \mathcal{D})$ are not continuous due the presence of the indicator functions. We thus compute the following continuous relaxation. Define the mean absolute activation $a_j = \mathbb{E}[|f_\theta(X)_j|]$ and its empirical version $\bar{a}_j = \frac{1}{n} \sum_{i=1}^{n} |f_\theta(x_i)_j|$, which is the $\ell_1$ norm of the activations (scaled by $1/n$) in contrast to the $\ell_0$ quasi norm in the FLOPs calculation. Define the relaxations, $\widetilde{\mathcal{F}}(f_\theta, \mathcal{P}) = \sum_{j=1}^{d} a_j^2$ and its consistent estimator $\widetilde{\mathcal{F}}(f_\theta, \mathcal{D}) = \sum_{j=1}^{d} \bar{a}_j^2$. We propose to minimize the following relaxation, which can be optimized using any off-the-shelf stochastic gradient

descent optimizer.

$$\min_{\theta \in \Theta} \underbrace{\ell(f_\theta, \mathcal{D}) + \lambda \widetilde{\mathcal{F}}(f_\theta, \mathcal{D})}_{\widetilde{\mathcal{L}}(\theta)}. \tag{4}$$

**Sparse retrieval and re-ranking.** During inference, the sparse vector of a query image is first obtained from the learned model and the nearest neighbour is searched in a database of sparse vectors forming a sparse matrix. An efficient algorithm to compute the dot product of the sparse query vector with the sparse matrix is presented in Figure 1. This consists of first building a list of the non-zero values and their positions in each column. As motivated in Section 3, given a sparse query vector, it is sufficient to only iterate through the non-zero values and the corresponding columns. Next, a filtering step is performed keeping only scores greater than a specified threshold. Top-$k$ candidates from the remaining items are returned. The complete algorithm is presented in Algorithm 1. In practice, the sparse retrieval step is not sufficient to ensure good performance. The top-$k$ shortlisted candidates are therefore further re-ranked using dense embeddings as done in SDH. This step involves multiplication of a small dense matrix with a dense vector. The number of shortlisted candidates $k$ is chosen such that the dense re-ranking time does not dominate the total time.

**Comparison to SDH (Jeong and Song, 2018).** It is instructive to contrast our approach with that of SDH (Jeong and Song, 2018). In contrast to the binary hashes in SDH, our approach learns sparse real valued representations. SDH uses a min-cost-max-flow approach in one of the training steps, while we train ours only using SGD. During inference in SDH, a shortlist of candidates is first created by considering the examples in the database that have hashes with non-empty intersections with the query hash. The candidates are further re-ranked using the dense embeddings. The shortlist in our approach on the other hand is constituted of the examples with the top scores from the sparse embeddings.

**Comparison to unrelaxed FLOPs regularizer.** We provide an experimental comparison of our continuous relaxation based FLOPs regularizer to its unrelaxed variant, showing that the performance of the two are markedly similar. Setting up this experiment requires some analytical simplifications based on recent deep neural network analyses. We first recall recent results that indicate that the output of a batch norm layer nearly follows a Gaussian distribution (Santurkar et al., 2018), so that in our context, we could make the simplifying approximation that $f_\theta(X)_j$ (where $X \sim \mathcal{P}$) is distributed as $\rho(Y)$ where $Y \sim \mathcal{N}(\mu_j(\theta), \sigma_j^2(\theta))$, $\rho$ is the ReLU activation used at the neural network output. We have modelled the pre-activation as a Gaussian distribution with mean and variance depending on the model parameters $\theta$. We experimentally verify that this assumption holds by minimizing the KS distance (Massey Jr, 1951) between the CDF of $\rho(Y)$ where $Y \sim \mathcal{N}(\mu, \sigma^2)$ and the empirical CDF of the activations. The KS distance is minimized wrt. $\mu, \sigma$. Figure 3a shows the empirical CDF and the fitted CDF of $\rho(Y)$ for two different architectures.

While $\mu_j(\theta), \sigma_j(\theta)$ ($1 \leq j \leq d$) cannot be tuned independently due to their dependence on $\theta$, in practice, the huge representational capacity of neural networks allows $\mu_j(\theta)$ and $\sigma_j(\theta)$ to be tuned almost independently. We consider a toy setting with 2-d embeddings. For a tractable analysis, we make the simplifying assumption that, for $j = 1, 2$, $f_\theta(X)_j$ is distributed as $\text{ReLU}(Y)$ where $Y \sim \mathcal{N}(\mu_j, \sigma_j^2)$, thus losing the dependence on $\theta$.

We now analyze how minimizing the continuous relaxation $\widetilde{\mathcal{F}}(f_\theta, \mathcal{P})$ compares to minimizing $\mathcal{F}(f_\theta, \mathcal{P})$. Note that we consider the population quantities here instead of the empirical quantities, as they are more amenable to theoretical analyses due to the existence of closed form expressions. We also consider the $\ell_1$ regularizer as a baseline. We initialize with $(\mu_1, \mu_2, \sigma_1, \sigma_2) = (-1/4, -1.3, 1, 1)$, and minimize the three quantities via gradient descent with infinitesimally small learning rates. For this contrastive analysis, we have not considered the effect of the metric loss. Note that while the discontinuous empirical quantity $\mathcal{F}(f_\theta, \mathcal{D})$ cannot be optimized via gradient descent, it is possible to do so for its population counterpart $\mathcal{F}(f_\theta, \mathcal{P})$ since it is available in closed form as a continuous function when making Gaussian assumptions. The details of computing the gradients can be found in Appendix A.

We start with activation probabilities $(p_1, p_2) = (0.4, 0.1)$, and plot the trajectory taken when performing gradient descent, shown in Figure 3b. Without the effect of the metric loss, the probabilities are expected to go to zero as observed in the plot. It can be seen that, in contrast to the $\ell_1$-regularizer,

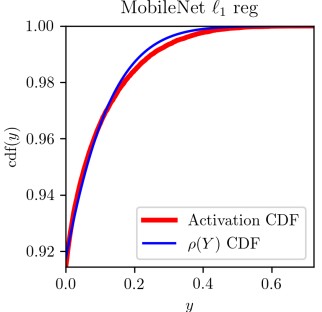 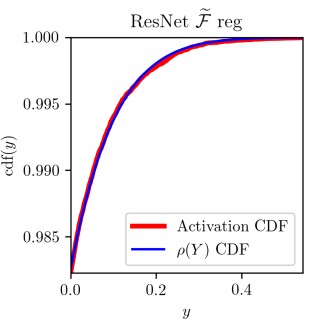 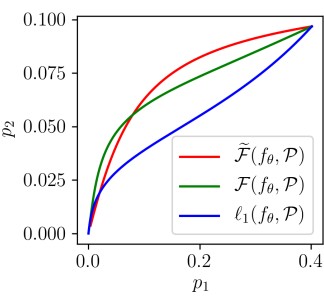

(a) The CDF of $\rho(Y)$ fitted to minimize the KS distance to the empirical CDF of the activations for two different architectures.

(b) The trajectory of the activation probabilities when minimizing the respective regularizers.

Figure 3: Figure (a) shows that the CDF of the activations (red) closely resembles the CDF of $\rho(Y)$ (blue) where $Y$ is a Gaussian random variable. Figure (b) shows that $\mathcal{F}$ and $\widetilde{\mathcal{F}}$ behave similarly by sparsifying the less sparser activation at a faster rate when compared to the $\ell_1$ regularizer.

$\mathcal{F}$ and $\widetilde{\mathcal{F}}$ both tend to sparsify the less sparse activation ($p_1$) at a faster rate, which *corroborates the fact that they encourage an even distribution of non-zeros.*

**$\widetilde{\mathcal{F}}$ promotes orthogonality.** We next show that, when the embeddings are normalized to have a unit norm, as typically done in metric learning, then minimizing $\widetilde{\mathcal{F}}(f_\theta, \mathcal{D})$ is equivalent to promoting orthogonality on the *absolute values* of the embedding vectors. Let $\|f_\theta(\boldsymbol{x})\|_2 = 1, \ \forall x \in \mathcal{X}$, we then have the following:

$$\widetilde{\mathcal{F}}(f_\theta, \mathcal{D}) = \sum_{j=1}^{d} \left( \frac{1}{n} \sum_{i=1}^{n} |f_\theta(\boldsymbol{x}_i)_j| \right)^2 = \frac{1}{n^2} \sum_{p,q \in [1:n]} \left\langle |f_\theta(\boldsymbol{x}_p)|, |f_\theta(\boldsymbol{x}_q)| \right\rangle \tag{5}$$

$\widetilde{\mathcal{F}}(f_\theta, \mathcal{D})$ is minimized when the vectors $\{|f_\theta(\boldsymbol{x}_i)|\}_{i=1}^{n}$ are orthogonal. Metric learning losses aim at minimizing the interclass dot product, whereas the FLOPs regularizer aims at minimizing pairwise dot products irrespective of the class, leading to a tradeoff between sparsity and accuracy. This approach of pushing the embeddings apart, bears some resemblance to the idea of *spreading vectors* (Sablayrolles et al., 2019) where an entropy based regularizer is used to uniformly distribute the embeddings on the unit sphere, albeit without considering any sparsity. Maximizing the pairwise dot product helps in reducing FLOPs as is illustrated by the following toy example. Consider a set of $d$ vectors $\{\boldsymbol{v}_i\}_{i=1}^{d} \subset \mathbb{R}^d$ (here $n = d$) satisfying $\|\boldsymbol{v}_i\|_2 = 1, \ \forall i \in [1:d]$. Then $\sum_{p,q \in [1:d]} \left\langle |\boldsymbol{v}_p|, |\boldsymbol{v}_q| \right\rangle$ is minimized when $\boldsymbol{v}_p = e_p$, where $e_p$ is an one-hot vector with the $p$ th entry equal to 1 and the rest 0. The FLOPs regularizer thus tends to spread out the non-zero activations in all the dimensions, thus producing balanced embeddings. This simple example also demonstrates that when the number of classes in the training set is smaller or equal to the number of dimensions $d$, a trivial embedding that minimizes the metric loss and also achieves a small number of FLOPs is $f_\theta(\boldsymbol{x}) = e_y$ where $y$ is true label for $\boldsymbol{x}$. This is equivalent to predicting the class of the input instance. The caveat with such embeddings is that they might not be semantically meaningful beyond the specific supervised task, and will naturally hurt performance on unseen classes, and tasks where the representation itself is of interest. In order to avoid such a collapse in our experiments, we ensure that the embedding dimension is smaller than the number of training classes. Furthermore, as recommended by Sablayrolles et al. (2017), we perform all our evaluations on unseen classes.

**Exclusive lasso.** Also known as $\ell_{1,2}$-norm, in previous works it has been used to induce competition (or exclusiveness) in features in the same group. More formally, consider $d$ features indexed by $\{1, \ldots, d\}$, and groups $g \subset \{1, \ldots, d\}$ forming a set of groups $\mathcal{G} \subset 2^{\{1,\ldots,d\}}$.[2] Let $\boldsymbol{w}$ denote the

---

[2]Denotes the powerset of $\{1, \ldots, d\}$.

weight vector for a linear classifier. The exclusive lasso regularizer is defined as,

$$\Omega_{\mathcal{G}}(\boldsymbol{w}) = \sum_{g \in \mathcal{G}} \|\boldsymbol{w}_g\|_1^2,$$

where $\boldsymbol{w}_g$ denotes the sub-vector $(\boldsymbol{w}_i)_{i \in g}$, corresponding to the indices in $g$. $\mathcal{G}$ can be used to induce various kinds of structural properties. For instance $\mathcal{G}$ can consist of groups of correlated features. The regularizer prevents feature redundancy by selecting only a few features from each group.

Our proposed FLOPs based regularizer has the same form as exclusive lasso. Therefore exclusive lasso applied to the batch of activations, with the groups being columns of the activation matrix (and rows corresponding to different inputs), is equivalent to the FLOPs regularizer. It can be said that, within each activation column, the FLOPs regularizer induces competition between different input examples for having a non-zero activation.

## 5 EXPERIMENTS

We evaluate our proposed approach on a large scale metric learning dataset: the Megaface (Kemelmacher-Shlizerman et al., 2016) used for face recognition. This is a much more fine grained retrieval tasks (with 85k classes for training) compared to the datasets used by Jeong and Song (2018). This dataset also satisfies our requirement of the number of classes being orders of magnitude higher than the dimensions of the sparse embedding. As discussed in Section 4, a few number of classes during training can lead the model to simply learn an encoding of the training classes and thus not generalize to unseen classes. Face recognition datasets avoid this situation by virtue of the huge number of training classes and a balanced distribution of examples across all the classes.

Following standard protocol for evaluation on the Megaface dataset (Kemelmacher-Shlizerman et al., 2016), we train on a refined version of the MSCeleb-1M (Guo et al., 2016) dataset released by Deng et al. (2018) consisting of 1 million images spanning 85k classes. We evaluate with 1 million distractors from the Megaface dataset and 3.5k query images from the Facescrub dataset (Ng and Winkler, 2014), which were not seen during training.

**Network architecture.** We experiment with two architectures: MobileFaceNet (Chen et al., 2018), and ResNet-101 (He et al., 2016). We use ReLU activations in the embedding layer for Mobile-FaceNet, and SThresh activations (defined below) for ResNet. The activations are $\ell_2$-normalized to produce an embedding on the unit sphere, and used to compute the Arcface loss (Deng et al., 2018). We learn 1024 dimensional sparse embeddings for the $\ell_1$ and $\widetilde{\mathcal{F}}$ regularizers; and 128, 512 dimensional dense embeddings as baselines. All models were implemented in Tensorflow (Abadi et al., 2016) with the sparse retrieval algorithm implemented in C++. The re-ranking step used 512-d dense embeddings.

**Activation function.** In practice, having a non-linear activation at the embedding layer is crucial for sparsification. Layers with activations such as ReLU are easier to sparsify due to the bias parameter in the layer before the activation (linear or batch norm) which acts as a direct control knob to the sparsity. More specifically, $\text{ReLU}(\boldsymbol{x} - \boldsymbol{\lambda})$ can be made more (less) sparse by increasing (decreasing) the components of $\boldsymbol{\lambda}$, where $\boldsymbol{\lambda}$ is the bias parameter of the previous linear layer. In this paper we consider two types of activations: $\text{ReLU}(\boldsymbol{x}) = \max(\boldsymbol{x}, \boldsymbol{0})$, and the soft thresholding operator $\text{SThresh}(\boldsymbol{x}) = \text{sgn}(\boldsymbol{x}) \max(|\boldsymbol{x}| - 1/2, 0)$ (Boyd and Vandenberghe, 2004). ReLU activations always produce positive values, whereas soft thresholding can produce negative values as well.

**Practical considerations.** In practice, setting a large regularization weight $\lambda$ from the beginning is harmful for training. Sparsifying too quickly using a large $\lambda$ leads to many dead activations (saturated to zero) in the embedding layer and the model getting stuck in a local minima. Therefore, we use an annealing procedure and gradually increase $\lambda$ throughout the training using a regularization weight schedule $\lambda(t) : \mathbb{N} \mapsto \mathbb{R}$ that maps the training step to a real valued regularization weight. In our experiments we choose a $\lambda(t)$ that increases quadratically as $\lambda(t) = \lambda(t/T)^2$, until step $t = T$, where $T$ is the threshold step beyond which $\lambda(t) = \lambda$.

**Baselines.** We compare our proposed $\widetilde{\mathcal{F}}$-regularizer, with multiple baselines: exhaustive search with dense embeddings, sparse embeddings using $\ell_1$ regularization, *Sparse Deep Hashing (SDH)*

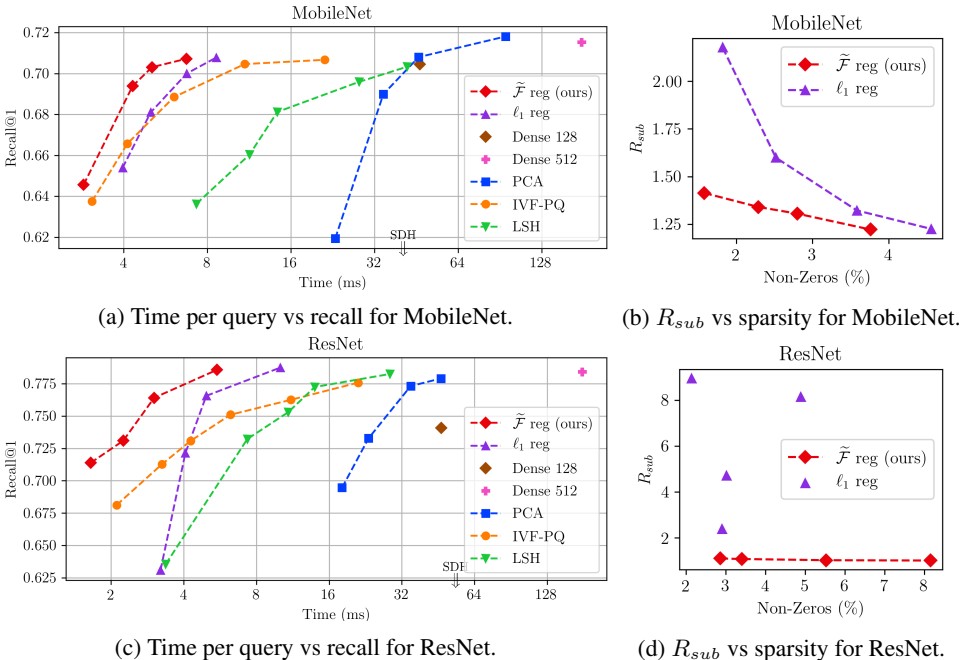

Figure 4: Figures (a) and (c) show the speed vs recall trade-off for the MobileNet and ResNet architectures respectively. The trade-off curves produced by varying the hyper-parameters of the respective approaches. The points with higher recall and lower time (top-left side of the plots) are better. The SDH baseline being out of range of both the plots is indicated using an arrow. Figures (b) and (d) show the sub-optimality ratio vs sparsity plots for MobileNet and ResNet respectively. $R_{sub}$ closer to 1 indicates that the non-zeros are uniformly distributed across the dimensions.

(Jeong and Song, 2018), and PCA, LSH, PQ applied to the 512 dimensional dense embeddings from both the architectures. We train the SDH model using the aforementioned architectures for 512 dimensional embeddings, with number of active hash bits $k = 3$. We use numpy (using efficient MKL optimizations in the backend) for matrix multiplication required for exhaustive search in the dense and PCA baselines. We use the CPU version of the *Faiss* (Johnson et al., 2017) library for LSH and PQ (we use the IVF-PQ index from Faiss).

Further details on the training hyperparameters and the hardware used can be found in Appendix B.

## 5.1 RESULTS

We report the recall and the time-per-query for various hyperparameters of our proposed approach and the baselines, yielding trade-off curves. The reported times include the time required for re-ranking. The trade-off curves for MobileNet and ResNet are shown in Figures 4a and 4c respectively. We observe that while vanilla $\ell_1$ regularization is an improvement by itself for some hyperparameter settings, the $\widetilde{\mathcal{F}}$ regularizer is a further improvement, and yields the most optimal trade-off curve. SDH has a very poor speed-accuracy trade-off, which is mainly due to the explosion in the number of shortlisted candidates with increasing number of active bits leading to an increase in the retrieval time. On the other hand, while having a small number of active bits is faster, it leads to a smaller recall. For the other baselines we notice the usual order of performance, with PQ having the best speed-up compared to LSH and PCA. While dimensionality reduction using PCA leads to some speed-up for relatively high dimensions, it quickly wanes off as the dimension is reduced even further.

We also report the sub-optimality ratio $R_{sub} = \mathcal{F}(f_\theta, \mathcal{D})/d\bar{p}^2$ computed over the dataset $\mathcal{D}$, where $\bar{p} = \frac{1}{d}\sum_{j=1}^{d} \bar{p}_j$ is the mean activation probability estimated on the test data. Notice that $R_{sub} \geq 1$, and the optimal $R_{sub} = 1$ is achieved when $\bar{p}_j = \bar{p}, \forall 1 \leq j \leq d$, that is when the non-zeros are evenly distributed across the dimensions. The sparsity-vs-suboptimality plots for MobileNet and ResNet are shown in Figures 4a and 4c respectively. We notice that the $\widetilde{\mathcal{F}}$-regularizer yields values of

$R_{sub}$ closer to 1 when compared to the $\ell_1$-regularizer. For the MobileNet architecture we notice that the $\ell_1$ regularizer is able to achieve values of $R$ close to that of $\widetilde{\mathcal{F}}$ in the less sparser region. However, the gap increases substantially with increasing sparsity. For the ResNet architecture on the other hand the $\ell_1$ regularizer yields extremely sub-optimal embeddings in all regimes. The $\widetilde{\mathcal{F}}$ regularizer is therefore able to produce more balanced distribution of non-zeros.

The sub-optimality is also reflected in the recall values. The gap in the recall values of the $\ell_1$ and $\widetilde{\mathcal{F}}$ models is much higher when the sub-optimality gap is higher, as in the case of ResNet, while it is small when the sub-optimality gap is smaller as in the case of MobileNet. This shows the significance of having a balanced distribution of non-zeros. Additional results, including results without the re-ranking step and performance on CIFAR-100 can be found in Appendix C.

## 6 CONCLUSION

In this paper we proposed a novel approach to learn high dimensional embeddings with the goal of improving efficiency of retrieval tasks. Our approach integrates the FLOPs incurred during retrieval into the loss function as a regularizer and optimizes it directly through a continuous relaxation. We provide further insight into our approach by showing that the proposed approach favors an even distribution of the non-zero activations across all the dimensions. We experimentally showed that our approach indeed leads to a more even distribution when compared to the $\ell_1$ regularizer. We compared our approach to a number of other baselines and showed that it has a better speed-vs-accuracy trade-off. Overall we were able to show that sparse embeddings can be around $50\times$ faster compared to dense embeddings without a significant loss of accuracy.

**Acknowledgements**
We thank Hong-You Chen for helping in running some baselines during the early stages of this work. This work has been partially funded by the DARPA D3M program and the Toyota Research Institute. Toyota Research Institute ("TRI") provided funds to assist the authors with their research but this article solely reflects the opinions and conclusions of its authors and not TRI or any other Toyota entity.

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

# Appendix

## A  GRADIENT COMPUTATIONS FOR ANALYTICAL EXPERIMENTS

As described in the main text, for purposes of an analytical toy experiment, we consider a simplified setting with 2-d embeddings with the $j$th ($j = 1, 2$) activation being distributed as $(Y_j)_+ = \text{ReLU}(Y_j)$ where $Y_j \sim \mathcal{N}(\mu_j, \sigma_j)$. We assume $\mu_j \leq 0$, which is typical for sparse activations ($p_j \leq 0.5$). Then the three compared regularizers are $\mathcal{F}(p_\theta, \mathcal{P}) = \sum_{j=1}^2 \mathbb{P}((Y_j)_+ > 0)^2$, $\widetilde{\mathcal{F}}(p_\theta, \mathcal{P}) = \sum_{j=1}^2 \mathbb{E}[(Y_j)_+]^2$, and $\ell_1(p_\theta, \mathcal{P}) = \sum_{j=1}^2 \mathbb{E}[(Y_j)_+]$. Computing the regularizer gradients thus boils down to computing the gradients of $\mathbb{P}((Y_j)_+ > 0)^2, \mathbb{E}[(Y_j)_+]^2$, and $\mathbb{E}[(Y_j)_+]$ as provided in the following lemmas. We hide the subscript $j$ for brevity, as computations are similar for all $j$.

**Lemma 1.**
$$\mathbb{E}[Y_+] = \frac{\sigma}{\sqrt{2\pi}} \exp\left(-\frac{\mu^2}{2\sigma^2}\right) + \mu\left(1 - \Phi\left(-\frac{\mu}{\sigma}\right)\right), \tag{6}$$

*and,*
$$\mathbb{P}(Y_+ > 0) = 1 - \Phi\left(-\frac{\mu}{\sigma}\right), \tag{7}$$

*where $\Phi$ denotes the cdf of the Gaussian distribution.*

*Proof of Lemma 1.* The proof is based on standard Gaussian identities.

$$\mathbb{E}[Y_+] = \int_0^\infty \frac{x}{\sqrt{2\pi\sigma^2}} \exp\left(-\frac{(x-\mu)^2}{2\sigma^2}\right) dx = \int_{-\mu}^\infty \frac{x+\mu}{\sqrt{2\pi\sigma^2}} \exp\left(-\frac{x^2}{2\sigma^2}\right) dx$$

$$= \int_{-\mu}^\infty \frac{x}{\sqrt{2\pi\sigma^2}} \exp\left(-\frac{x^2}{2\sigma^2}\right) dx + \int_{-\mu}^\infty \frac{\mu}{\sqrt{2\pi\sigma^2}} \exp\left(-\frac{x^2}{2\sigma^2}\right) dx$$

$$= \frac{\sigma}{\sqrt{2\pi}} \exp\left(-\frac{\mu^2}{2\sigma^2}\right) + \mu\left(1 - \Phi\left(-\frac{\mu}{\sigma}\right)\right)$$

$$\mathbb{P}(Y_j > 0) = \int_0^\infty \frac{1}{\sqrt{2\pi\sigma^2}} \exp\left(-\frac{(x-\mu)^2}{2\sigma^2}\right) dx = \int_{-\mu/\sigma}^\infty \frac{1}{\sqrt{2\pi}} \exp\left(-\frac{x^2}{2}\right) dx$$

$$= 1 - \Phi\left(-\frac{\mu}{\sigma}\right)$$

$\square$

**Lemma 2.**
$$\frac{\partial}{\partial\mu} \mathbb{P}(Y_+ > 0) = -\frac{\partial}{\partial\mu} \Phi\left(-\frac{\mu}{\sigma}\right) = \frac{1}{\sigma\sqrt{2\pi}} \exp\left(-\frac{\mu^2}{2\sigma^2}\right). \tag{8}$$

$$\frac{\partial}{\partial\sigma} \mathbb{P}(Y_+ > 0) = -\frac{\partial}{\partial\sigma} \Phi\left(-\frac{\mu}{\sigma}\right) = -\frac{\mu}{\sigma^2\sqrt{2\pi}} \exp\left(-\frac{\mu^2}{2\sigma^2}\right). \tag{9}$$

*Proof of Lemma 2.* Follows directly from the statement by standard differentiation.  $\square$

**Lemma 3.**
$$\frac{\partial}{\partial\mu} \mathbb{E}[Y_+] = 1 - \Phi\left(-\frac{\mu}{\sigma}\right). \tag{10}$$

$$\frac{\partial}{\partial\sigma} \mathbb{E}[Y_+] = \frac{1}{\sqrt{2\pi}} \exp\left(-\frac{\mu^2}{2\sigma^2}\right). \tag{11}$$

*Proof of Lemma 3.*
$$\frac{\partial}{\partial\mu} \mathbb{E}[Y_+] = -\frac{\mu}{\sigma\sqrt{2\pi}} \exp\left(-\frac{\mu^2}{2\sigma^2}\right) + \frac{\partial}{\partial\mu}\left[\mu\left(1 - \Phi\left(-\frac{\mu}{\sigma}\right)\right)\right] = 1 - \Phi\left(-\frac{\mu}{\sigma}\right)$$

where the last step follows from Lemma 2.

$$\frac{\partial}{\partial \sigma} \mathbb{E}[Y_+] = \frac{1}{\sqrt{2\pi}} \exp\left(-\frac{\mu^2}{2\sigma^2}\right) + \frac{\mu^2}{\sigma^2 \sqrt{2\pi}} \exp\left(-\frac{\mu^2}{2\sigma^2}\right) + \frac{\partial}{\partial \sigma}\left[\mu\left(1 - \Phi\left(-\frac{\mu}{\sigma}\right)\right)\right]$$
$$= \frac{1}{\sqrt{2\pi}} \exp\left(-\frac{\mu^2}{2\sigma^2}\right)$$

where the last step follows from Lemma 2. □

**Lemma 4.**

$$\frac{\partial}{\partial \mu} \mathbb{E}[Y_+]^2 = 2\mathbb{E}[Y_+]\left(1 - \Phi\left(-\frac{\mu}{\sigma}\right)\right). \tag{12}$$

$$\frac{\partial}{\partial \sigma} \mathbb{E}[Y_+]^2 = 2\mathbb{E}[Y_+]\frac{1}{\sqrt{2\pi}} \exp\left(-\frac{\mu^2}{2\sigma^2}\right). \tag{13}$$

*Proof of Lemma 4.* Follows directly from Lemma 3. □

**Lemma 5.**

$$\frac{\partial}{\partial \mu} \mathbb{P}(Y_+ > 0)^2 = 2\mathbb{P}(Y_+ > 0)\frac{1}{\sigma\sqrt{2\pi}} \exp\left(-\frac{\mu^2}{2\sigma^2}\right). \tag{14}$$

$$\frac{\partial}{\partial \sigma} \mathbb{P}(Y_+ > 0)^2 = -2\mathbb{P}(Y_+ > 0)\frac{\mu}{\sigma^2\sqrt{2\pi}} \exp\left(-\frac{\mu^2}{2\sigma^2}\right). \tag{15}$$

*Proof of Lemma 5.* Follows directly from Lemma 2. □

# B EXPERIMENTAL DETAILS

All images were resized to size $112 \times 112$ and aligned using a pre-trained aligner[3]. For the Arcloss function, we used the recommended parameters of margin $m = 0.5$ and temperature $s = 64$. We trained our models on 4 NVIDIA Tesla V-100 GPUs using SGD with a learning rate of $0.001$, momentum of $0.9$. Both the architectures were trained for a total of 230k steps, with the learning rate being decayed by a factor of 10 after 170k steps. We use a batch size of 256 and 64 per GPU for MobileFaceNet for ResNet respectively.

Pre-training in SDH is performed in the same way as described above. The hash learning step is trained on a single GPU with a learning rate of $0.001$. The ResNet model is trained for 200k steps with a batch size of 64, and the MobileFaceNet model is trained for 150k steps with a batch size of 256. We set the number of active bits $k = 3$ and a pairwise cost of $p = 0.1$.

**Hyper-parameters for MobileNet models.**

1. The regularization parameter $\lambda$ for the $\widetilde{\mathcal{F}}$ regularizer was varied as 200, 300, 400, 600.

2. The regularization parameter $\lambda$ for the $\ell_1$ regularizer was varied as 1.5, 2.0, 2.7, 3.5.

3. The PCA dimension is varied as 64, 96, 128, 256.

4. The number of LSH bits were varied as 512, 768, 1024, 2048, 3072.

5. For IVF-PQ from the faiss library, the following parameters were fixed: `nlist=4096, M=64, nbit=8`, and `nprobe` was varied as 100, 150, 250, 500, 1000.

---

[3]`https://github.com/deepinsight/insightface`

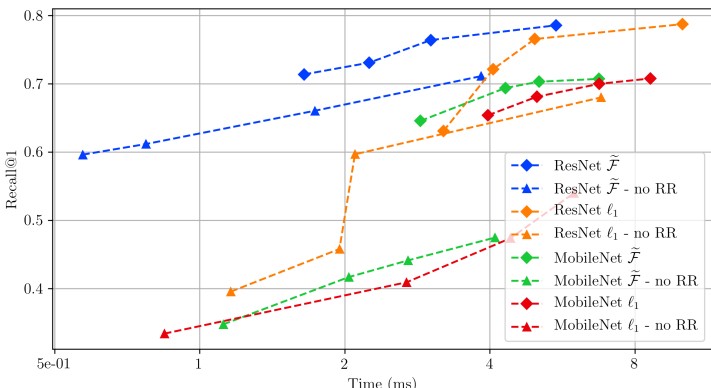

Figure 5: Time vs Recall@1 plots for retrieval with and without re-ranking. Results from the same model and regularizer have same colors. Diamonds ($\Diamond$) denote results with re-ranking, and triangles ($\triangle$) denote results without re-ranking.

**Hyper-parameters for ResNet baselines.**

1. The regularization parameter $\lambda$ for the $\widetilde{\mathcal{F}}$ regularizer was varied as 50, 100, 200, 630.
2. The regularization parameter $\lambda$ for the $\ell_1$ regularizer was varied as 2.0, 3.0, 5.0, 6.0.
3. The PCA dimension is varied as 48, 64, 96, 128.
4. The number of LSH bits were varied as 256, 512, 768, 1024, 2048.
5. For IVF-PQ, the following parameters were the same as in MobileNet: `nlist=4096, M=64, nbit=8. nprobe` was varied as 50, 100, 150, 250, 500, 1000.

**Selecting top-$k$.** We use the following heuristic to create the shortlist of candidates after the sparse ranking step. We first shortlist all candidates with a score greater than some confidence threshold. For our experiments we set the confidence threshold to be equal to 0.25. If the size of this shortlist is larger than $k$, it is further shrunk by consider the top $k$ scorers. For all our experiments we set $k = 1000$. This heuristic avoids sorting the whole array, which can be a bottleneck in this case. The parameters are chosen such that the time required for the re-ranking step does not dominate the total retrieval time.

**Hardware.**

1. All models were trained on 4 NVIDIA Tesla V-100 GPUs with 16G of memory.
2. System Memory: 256G.
3. CPU: Intel(R) Xeon(R) CPU E5-2686 v4 @ 2.30GHz.
4. Number of threads: 32.
5. Cache: L1d cache 32K, L1i cache 32K, L2 cache 256K, L3 cache 46080K.

All timing experiments were performed on a single thread in isolation.

## C  ADDITIONAL RESULTS

### C.1  RESULTS WITHOUT RE-RANKING

Figure 5 shows the comparison of the approaches with and without re-ranking. We notice that there is a significant dip in the performance without re-ranking with the gap being smaller for ResNet with FLOPs regularization. We also notice that the FLOPs regularizers has a better trade-off curve for the no re-ranking setting as well.

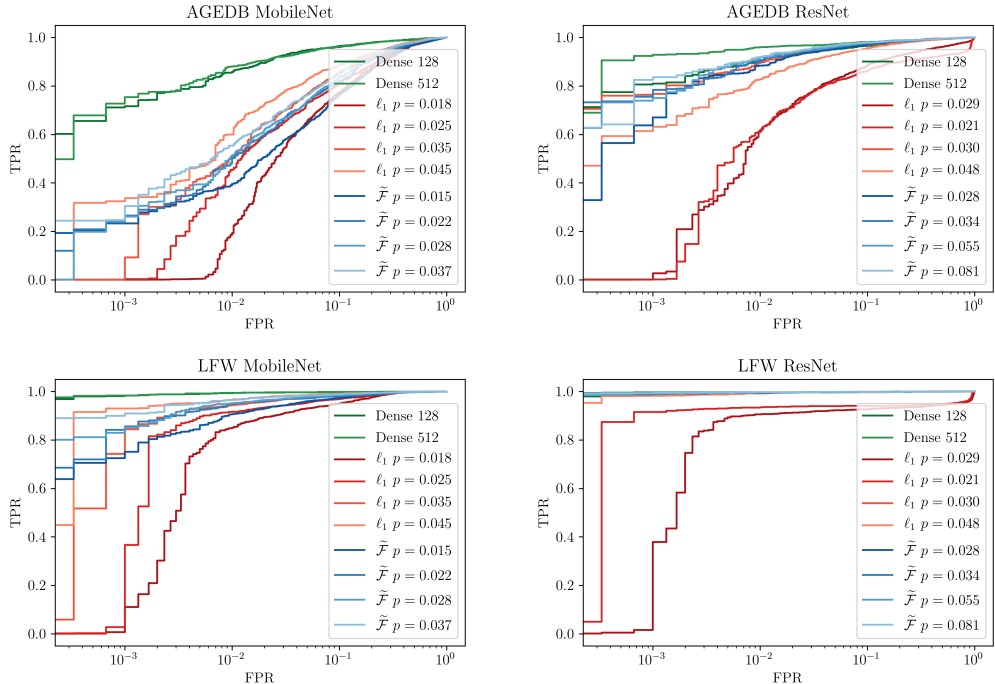

Figure 6: FPR-TPR curves. The $\ell_1$ curves are all shown in shades of red, where as the FLOPs curves are all shown in shades of blue. The probability of activation is provided in the legend for comparison. For curves with similar probability of activation $p$, the FLOPs regularizer performs better compared to $\ell_1$, thus demonstrating that the FLOPs regularizer learns richer representations for the same sparsity.

## C.2   FPR AND TPR CURVES

In the main text we have reported the recall@1 which is a standard face *recognition* metric. This however is not sufficient to ensure good face *verification* performance. The goal in face verification is to predict whether two faces are similar or dissimilar. A natural metric in such a scenario is the FPR-TPR curve. Standard face verification datasets include LFW (Huang et al., 2008) and AgeDB (Moschoglou et al., 2017). We produce embeddings using our trained models, and use them to compute similarity scores (dot product) for pairs of images. The similarity scores are used to compute the FPR-TPR curves which are shown in Figure 6. We notice that for curves with similar probability of activation $p$, the FLOPs regularizer performs better compared to $\ell_1$. This demonstrates the efficient utilization of all the dimensions in the case of the FLOPs regularizer that helps in learning richer representations for the same sparsity.

We also observe that the gap between sparse and dense models is smaller for ResNet, thus suggesting that the ResNet model learns better representations due to increased model capacity. Lastly, we also note that the gap between the dense and sparse models is smaller for LFW compared to AgeDB, thus corroborating the general consensus that LFW is a relatively easier dataset.

## C.3   CIFAR-100 RESULTS

We also experimented with the Cifar-100 dataset (Krizhevsky et al., 2009) consisting of 60000 examples and 100 classes. Each class consists of 500 train and 100 test examples. We compare the $\ell_1$ and FLOPs regularized approaches with the sparse deep hashing approach. All models were trained using the triplet loss (Schroff et al., 2015) and embedding dim $d = 64$. For the dense and DH baselines, no activation was used on the embeddings. For the $\ell_1$ and FLOPs regularized models we used the SThresh activation. Similar to Jeong and Song (2018), the train-test and test-test precision values have been reported in Table 1. Furthermore, the reported results are without re-ranking. Cifar-100 being a small dataset, we only report the FLOPs-per-row, as time measurements can be

| Model | $F$ | Train | | Test | |
|---|---|---|---|---|---|
| | | prec@4 | prec@16 | prec@4 | prec@16 |
| Dense | 64 | 61.53 | 61.26 | **57.31** | **56.95** |
| SDH $k = 1$ | **1.18** | **62.29** | **61.94** | 57.22 | 55.87 |
| SDH $k = 2$ | 3.95 | 60.93 | 60.15 | 55.98 | 54.42 |
| SDH $k = 3$ | 8.82 | 60.80 | 59.96 | 55.81 | 54.10 |
| $\widetilde{F}$ no re-ranking | **0.40** | **61.05** | **61.08** | 55.23 | **55.21** |
| $\ell_1$ no re-ranking | 0.47 | 60.50 | 60.17 | 54.32 | 54.96 |

Table 1: Cifar-100 results using triplet loss and embedding size $d = 64$. For $\ell_1$ and $\widetilde{\mathcal{F}}$ models, no re-ranking was used. $F$ is used to denote the FLOPs-per-row (lower is better). The SDH results have been reported from the original paper.

misleading. In our experiments, we achieved slightly higher precisions for the dense model compared to (Jeong and Song, 2018). We notice that our models use less than $50\%$ of the computation compared to SDH, albeit with a slightly lower precision.

