# OpenReview forum: "Minimizing FLOPs to Learn Efficient Sparse Representations"
_ICLR.cc/2020/Conference — Accept (Poster)_

### Official Review · AnonReviewer4 · 2019-10-22
**Official Blind Review #4**

**Rating:** 8

**Review:**

Summary

This paper focuses on learning a representation that facilitates efficient content-based retrieval. Although the representations that are learned from deep neural networks can contain rich information, it is computationally expensive to use those representations to perform a search for the best match. In particular, computing the Euclidean distance between a query and an instance scales linearly with the size of the representation. Prior approaches to this problem have focused either on: (1) compactifying the learned representations into another form, such as a Hamming code, in a way that preserves the identifiability of an instance; (2) resorting to approximate methods that sacrifice accurate search for efficiency.

To address these inconvenient trade-offs, this paper proposes an algorithm to learn a high-dimensional, sparse representation that is directly used in retrieval, instead of learning a separate, more compact representation. The novelty of this algorithm is its focus on minimizing the number of FLOPs in computing queries of instances, taking note as well of the role of the distribution of non-zero values in determining the number of FLOPs. A continuous relaxation of the equations thus derived provides the proposed algorithm. The paper notes differences between the algorithm and SDH, a recent candidate that learns a sparse, high-high dimensional hash. Based on experiments, the paper claims that the proposed algorithm yields a similar or better speed vs. recall tradeoff compared to baselines. The paper also provides additional experiments demonstrating the sparsity of the representations and the even distribution of non-zero values of the proposed regularizer.

Decision: accept

The algorithm is clearly and succinctly motivated from the standpoint of reducing the number of FLOPs. The presentation of the distribution that minimizes FLOPs is convincing, and there is easy-to-follow buildup into the continuous relaxation of the FLOPs minimization problem.

The proposed algorithm is itself relatively simple (just an additional regularizer term that can be optimized with any SGD-based optimizer), compared to other methods that learn a separate representation or that use approximate nearest-neighbour search, and directly tries to address aforementioned trade-offs between efficiency of retrieval and richness of the representation.

I found helpful the comparison both to the nearest competitor method (SDH) and the unrelaxed regularizer. The additional experiments comparing the continuous relaxation and the unrelaxed regularizer were interesting, but I found Figure 2b a little hard to understand. I also appreciated the intuition developed at the end of section 4 for how the regularizer promotos orthogonality.

The recall/time trade-off curves in figure 3 support the main empirical claim of the paper. The sparsity plots in figure 3 contributed to a broader understanding of the algorithm besides based on metrics other than accuracy.

Nevertheless, there were some experimental presentation issues. Are errors bars possible for figs 3ac? Some evaluation metrics that are present in other papers are also missing, which might provide a more complete understanding of algorithm's advantages and disadvantages. Specifically, precision@k (Jeong and Song, 2018) and true-positive/false-positive curves (Kemelmacher-Shlizerman et. al., 2016) could help readers in making an informed algorithmic choice. It also seems that the paper could have included more comparisons to other ways of learning sparse representations (e.g., L2, top-k autoencoders, dropout), and included a broader literature review of learning sparse representations (see here: https://arxiv.org/pdf/1505.05561.pdf).

Questions
1. Did confidence thresholds in ranking (Appendix B, re-ranking) affect the results in any way? Did it depend on the algorithm used?

Minor comments that did not affect the rating
1. Typo in figure 3; the second sentence should say "curves are produced by"
2. Typos in Appendix A
a. Lemmas 2, 5 should have X_+ instead of X in the statements
b. Second last line of proof of lemma 1 should have X_+
3. Would be helpful to have whole procedure from learning to retrieval in an algorithm box
4. Include a summary of the problem setting before section 3. It is at first a bit unclear what the problem is for newcomers
5. Add algorithm header to the retrieval algorithm presented in fig 1




**Experience Assessment:**

I have read many papers in this area.

**Review Assessment: Checking Correctness Of Derivations And Theory:**

I carefully checked the derivations and theory.

**Review Assessment: Checking Correctness Of Experiments:**

I assessed the sensibility of the experiments.

**Review Assessment: Thoroughness In Paper Reading:**

I read the paper thoroughly.

---

> ### Author Response · Authors · 2019-11-11
> **Response to review #4**
>
> Thank you for the feedback and suggestions. We have now added some more discussion on figure 2b in section 4. Responses to other issues:
>
> 1. Literature review on learning sparse representations: Thank you for pointing us to the literature on sparse AE. We have now expanded the literature review with approaches for learning sparse representations.
>
> 2. More metrics: we have now added TPR-FPR curves in appendix C of our first revision. Our plots show that for similar probability of activation $p$, the FLOPs regularizer performs better compared to $\ell_1$. This demonstrates the efficient utilization of all the dimensions in the case of the FLOPs regularizer that helps in learning richer representations for the same sparsity.
>
> 3. Confidence thresholds: Both the evaluation metric and the retrieval time are not so sensitive to the choice of the confidence threshold. The threshold is chosen such that the size of the shortlist is large enough (around 1000 in our experiments) so that it contains the top-k examples (where k is what appears in recall@k ), so as to not affect the metric. Also the chosen threshold is small enough that the time is dominated by the sparse multiplication time, so it does not significantly affect the total time either. Without a threshold however, the time for sorting the scores dominate, resulting in poor speedup.

---

> ### Author Response · Authors · 2019-11-15
> **Further updates**
>
> Further updates:
>
> 4. We have now added results on the Cifar-100 dataset in Appendix C, reporting the precision. Our results indicate that our models use less than $50\%$ computation compared to SDH, however with a slightly lower precision.

---

### Official Review · AnonReviewer3 · 2019-10-23
**Official Blind Review #3**

**Rating:** 3

**Review:**

This paper proposes to learn sparse representation in neural networks for retrieval in large database of vectors. Such sparse representation, when the fraction of non-zeros is high, can be computed using sparse matrix multiplication, or variants of inverted index scoring and lead to potentially lower FLOPs needed. This paper proposes to induce sparsity by adding a regularization term, which counts the expected number of FLOPs needed for sparse scoring.

The final experiments were done on Megaface dataset, where there are 1M distrators and accuracy is measured in Recall@1. The sparse embedding approach is combined with a dense re-ranking stage, and the speed-recall trade is compared to the pure dense approaches (such as using baselines such as FAISS’s IVF-PQ, LSH or SDH). The authors’ evaluation showed that, the performance of FLOPs regularized embedding is better than L1 regularized sparse embedding, or applying IVF-PQ directly to dense embeddings.

Pros:
- The topic of learning sparse embeddings is of great interests. As far as I know, many researchers attempted and there was not an agreement. For example, https://arxiv.org/pdf/1904.10631.pdf [1] claimed "Using sparse operations reduces the number of FLOPs, as the zero entries can be ignored. However, in general, sparsity leads to irregular memory access patterns, so it may not actually result in a speedup;" This has been my experience as well. The authors’ embedding seems to be much more sparse (<5% non-zeros) than the sparse embeddings from other approaches and thus works better.

- The formulation is simple and intuitive - it appears easy enough to plug into a variety of embedding architectures (potentially all embeddings, NLP, CV, Audio).

Cons:
- The learned sparse embedding system still requires training a separate dense embedding for re-ranking, so essentially is a hybrid approach. One cannot simply "get rid of the dense". Ideally one would hope to not need two inference runs (for sparse and dense) and keep two database (sparse and dense). Maybe the author can report how sparse embedding performs on its own.

- It is widely known that inverted index is not FLOPs bound, as its FLOPs utilization in inverted index is typically low. Inverted index is almost always dominated by random memory accesses and thus ideally the regularizer should be modelling after cache miss / memory access pattern instead of FLOPs. I’d like to see authors gave more discussion on these topics instead of taking a FLOPs centric view (which is not true for inverted index).

- For comparison dense ANN, Faiss’s IVF-PQ is a relatively dated pipeline. It would be good to see how what the curve would look like for other dense technique such as HNSW [2] which performs better than IVF-PQ on http://ann-benchmarks.com/. Also dense ANN can also greatly benefit from the use of batching, which is not considered for this paper.

- Finally, I recommend the authors perform additional experiments on other datasets. As the authors suggested, sometimes sparse embedding learning risks collapse to predicting the classes. The unseen face queries avoid this problem to some extent. But I still worry that the Megaface task somehow allow representation to be much more sparse than other NLP/CV tasks. For example, authors can try BERT tasks with sparse embedding. It would be much more convincing to see this approach generalizes across many tasks.

===
Overall, I think this is a nice paper which takes a good step to improve the effectiveness of sparse embeddings over existing methods such as simple L1 regularization. However, with the need of training separate dense embeddings, and the fact that experiments were conducted only on 1 tasks with relatively weak ANN baseline, I’d lean towards rejection.

[1] Low-Memory Neural Network Training: A Technical Report
[2] Efficient and robust approximate nearest neighbor search using Hierarchical Navigable Small World graphs


**Experience Assessment:**

I have published in this field for several years.

**Review Assessment: Checking Correctness Of Derivations And Theory:**

I carefully checked the derivations and theory.

**Review Assessment: Checking Correctness Of Experiments:**

I carefully checked the experiments.

**Review Assessment: Thoroughness In Paper Reading:**

I read the paper at least twice and used my best judgement in assessing the paper.

---

> ### Author Response · Authors · 2019-11-11
> **Response to review #3**
>
> Thank you for the feedback and suggestions to improve our paper. Response to comments:
>
> 1. Flops vs. speedup: We have now added a paragraph on cache efficiency. While FLOPs reduction is a reasonable measure of speedup on primitive processors of limited parallelization and cache memory. FLOPs is not an accurate measure of actual speedup when it comes to mainstream commercial processors such as Intel’s CPUs and Nvidia’sGPUs, as the latter have cache and SIMD (Single-Instruction Multiple Data) mechanisms highly optimized for dense matrix multiplication, while sparse matrix multiplication are inherently less tailored to their cache and SIMD design (Sohoni et al., 2019). On the other hand, there have been threads of research on hardwares with cache and parallelization tailored to sparse operations that show speedup proportional to the FLOPs reduction (Han et al., 2016; Parashar et al., 2017). Modeling the cache and other hardware aspects can potentially lead to better performance but less generality and is left to our future works.
>
> 2. Need both dense and sparse embeddings: Our method does need both the embeddings to achieve good performance. The accuracy drops without the re-ranking step.
>
> 3. HNSW: One of the technical difficulties we face while evaluating using HNSW is that it does not support deleting elements. Deleting elements is essential for our particular evaluation framework. Our evaluation framework involves having only 1 facescrub target mixed with the megaface distractors in the database. Thus after testing each query-target pair, we require to delete the target from the database. Since deleting is not supported, the other option is to re-create the HNSW index each time, which is not a feasible option wrt the computation time. We are considering other evaluation metrics for which HNSW is a feasible option.

---

> ### Author Response · Authors · 2019-11-15
> **Further updates**
>
> Further updates:
>
> 4. We have now added results on the Cifar-100 dataset in Appendix C, reporting the precision without any re-ranking. Our results indicate that our models use less than $50\%$ of the computation compared to SDH,  however with a slightly lower precision.
>
> 5. We have also added results comparing reranking and no reranking in Appendix C. We notice that there is a significant dip in the performance without re-ranking with the gap being smaller for ResNet with FLOPs regularization. We also notice that the FLOPs regularizers has a better trade-off curve even for the no re-ranking setting.

---

### Official Review · AnonReviewer2 · 2019-11-02
**Official Blind Review #1**

**Rating:** 8

**Review:**

This paper reads well, with intriguing ideas that challenge the ‘dense’ approach to DNNs, excellent thought experiments and convincing experiments.

The reviewer is neither an expert in face verification and in K-NN retrieval algorithms but has a solid experience in sparse ML algorithms and group lasso algorithms.

In this respect, the algorithms proposed in this paper represent an excellent extension of existing sparse algorithms that go against the current trend of focusing on compact dense representations because this is what GPUs handle best.

Clarity: an excellent introduction (appreciated by a reviewer not up-to-date in the topic) introduces representation learning for retrieval, though says little about the sparse multiplication state-of-the-art or face verification.  The rest of the paper reads very well (I had to dig deep to find some clarifications in detailed comments). For lack of space, one has to through quite few references to fully understand the experiments.

Quality: there are only few equations in this paper, but they rely on excellent notation. The algorithm is also well formulated. What strikes me as very good are the ‘thought experiments’ that suggest (rather than prove) that the approach is well grounded: the end of section 4 is excellent

Originality and significance: working on sparse representations is quite ‘original’ now, especially as they are so GPU-unfriendly (I note the sparse algorithm is implemented in C++, and am looking forward to the code release). I hope the excellent results reported in this paper will incite others to revisit them. 15 years ago, such as paper would have been less original. The sparse vector sparse matrix product algorithm is well known and has been used in ML publications before, for instance:
-	Haffner, ICML 2006, “Fast Transpose Methods for Kernel Learning on Sparse Data”
-	Kudo and Matsumoto 2003 “Fast methods for kernel-based text analysis”
What seems to be original is the algorithm to balance sparsity probabilities. The reviewer is well aware that a single non sparse column can kill the performance of the sparse matrix multiplication algorithm, and this is probably the main reason this algorithm has not found broad usage. The derivation and the connection to group and exclusive Lasso are excellent.

Detailed comments:
-	page 5 “where we suppress the dependency of Mu and Sigma”. How do you do that?
-	Page 7 “The analysis in figure 2 follows similarly for Stresh”.  Probably more explanation is needed here. Replacing Relu with a sum of Relus over affine transforms of the Gaussian variable is a complex operation:  how does it keep the same curves?


**Experience Assessment:**

I have published one or two papers in this area.

**Review Assessment: Checking Correctness Of Derivations And Theory:**

I assessed the sensibility of the derivations and theory.

**Review Assessment: Checking Correctness Of Experiments:**

I assessed the sensibility of the experiments.

**Review Assessment: Thoroughness In Paper Reading:**

I read the paper at least twice and used my best judgement in assessing the paper.

---

> ### Author Response · Authors · 2019-11-11
> **Response to review #1**
>
> Thank you for the feedback, and pointing us to prior work using sparse computations. We have now added a paragraph on the literature for SpMV computations. Response to specific comments:
>
> 1. Dependency of mu and sigma: In practice mu and sigma depend on the activations produced by the dataset. The activations are further affected by the main loss function. For simplicity in this analysis, we only consider the effect of the regularizer and ignore the loss function. In such a setting, due to the representational capacity of the neural network, the mean and the variance can be controlled almost independently. Hence the simplifying assumption. In reality however, they are dependent on the dataset and the parameters of the network.
>
> 2. SThresh: Thanks for pointing this out. It is not necessary for SThresh to follow the same gradient pattern. We have now removed the line from the updated PDF.

---

### Author Response · Authors · 2019-11-11
**First revision for rebuttal**

We thank the reviewers for their very helpful feedback. We have incorporated some of the suggested changes in the first version of our revision. Here is a summary of the changes (also marked in blue in the updated pdf):

1. Added more literature on sparse multiplication approaches (section 2).
2. Added a broader review of learning sparse representations (section 2).
3. Added more discussion on FLOPs vs speedup (section 3).
4. Added more explanation on figure 2b in the analysis in section 4.
5. Added more results: TPR FPR curves (Appendix C).

Time permitting, we might be able to add some of the remaining suggestions before the rebuttal deadline.

---

### Author Response · Authors · 2019-11-15
**Second revision for rebuttal**

Here are the changes that we made in second revision before the rebuttal deadline:

1. A comparison of reranking vs no reranking has been added in Appendix C.
2. Added experiments on Cifar-100 dataset in Appendix C, reporting the precision.

---

### Decision · Program_Chairs · 2019-12-19

**Decision:**

Accept (Poster)

**Comment:**

This paper studies methods for using weight sparsification to reduce the computational load of network inference.  While there is not absolute consensus on whether this paper should be accepted, one of the main criticisms of this paper is that sparse compute is not always realistic or efficient on a GPU.  While this may be true of the current SOTA in hardware, emerging computing platforms and CPU libraries may handle sparse networks quite well.  For this reason, I am willing to down-weight this criticism. Based on the remaining comments, this paper has the merit to be accepted, even if it is a bit forward looking in terms of the hardware platforms it targets.